# Cost-utility analysis of robotic-assisted radical cystectomy for bladder cancer compared to open radical cystectomy in the United Kingdom

Felix Machleid[1,2,3], Jenessa Ho-Wrigley[1]*, Ameera Chowdhury[1], Anita Paliah[1], Ho Lam Poon[1], Elena Pizzo[4]

1 School of Public Health, Imperial College London, London, England, United Kingdom, 2 Charité-Universitätsmedizin Berlin, Corporate Member of Freie Universität Berlin, Humboldt-Universität zu Berlin, Berlin, Germany, 3 Berlin Institute of Health at Charité – Universitätsmedizin Berlin, BIH Biomedical Innovation Academy, BIH Charité (Junior) (Digital) Clinician Scientist Program, Berlin, Germany, 4 Department of Applied Health Research University College London, London, England, United Kingdom

* jenessanho@gmail.com

**Data Availability Statement:** All relevant data are within the paper and its Supporting information files.

## Abstract

### Background

Bladder cancer is the tenth most common cancer in the United Kingdom. Currently, open radical cystectomy (ORC) is the gold standard. Due to the risk of complications and a 2.3-8% mortality rate1, there is growing interest in the use of robot-assisted radical cystectomy (RARC). The aim of this study is to perform a cost-utility analysis, comparing RARC to ORC for bladder cancer patients from the perspective of the National Health Service England.

### Methods

A three-stage decision tree: surgery, post-surgery transfusions and complications, in a 90-day time horizon, was produced to simulate possible pathways of patients. The incremental cost-effectiveness ratio (ICER) was calculated based on data derived from current literature. Multiple univariate sensitivity analysis was carried out to evaluate influences of varying costs of RARC and ORC on the ICER.

### Results

The ICER for RARC compared to ORC resulted in £25,536/QALY. At the lower threshold of £20,000/QALY, RARC resulted in a negative NMB (£-4,843.32) and at the upper threshold of £30,000/QALY, a positive NMB (£624.61) compared to ORC. Threshold analysis showed that the intervention costs of £13,497 and £14,403 are met at the lower and upper threshold respectively. The univariate sensitivity analysis showed that the intervention costs of RARC or ORC, and the probabilities of complications, had the greatest impact on the ICER.

### Conclusion

As the resultant ICER did not fall below the £20,000/QALY threshold, our study did not provide a definitive recommendation for RARC for bladder cancer. Negative values for the

**Funding:** The authors received no specific funding for this work.

**Competing interests:** The authors have declared that no competing interests exist.

NMB at the lower threshold indicated the intervention was not feasible from a cost perspective. At the upper threshold of £30,000/QALY, this situation was reversed. The intervention became cost-effective. Therefore, further research is needed to justify the intervention.

## Introduction

Bladder cancer is the tenth most common cancer in the United Kingdom (UK) accounting for 3% of all cancer cases. It is 3–4 times more common in men than in women [1]. Main symptoms of bladder cancer include painless gross haematuria, irritative voiding symptoms and suprapubic or rectal pain [2]. Bladder cancer is classified based on how far it has spread to the bladder wall. It can be described as either non-muscle invasive or invasive [3].

As part of the primary treatment, 49% of patients require surgery to remove the tumour and 10% of patients require major resection surgery [3]. Its prevalence and nature of treatment make bladder cancer one of the most expensive cancers for the National Health Service (NHS) at £65 million annually [1]. So far, there have been few breakthroughs in treatment options and little improvement in life expectancy over the past 30 years [3].

Currently, in the UK, the National Institute for Health and Care Excellence (NICE) recommends radical cystectomy for patients with high-risk bladder cancer [4]. Radical cystectomy is often followed by urinary diversion with ileal conduit being the preferred method [5, 6]. Although open radical cystectomy (ORC) is currently considered the gold standard, there is growing interest in robotic-assisted radical cystectomy (RARC) as ORC has risks of substantial blood loss, complications (30–70%) and mortality (2.3–8%) [7–9]. RARC is a laparoscopic technique in which the arms of the robotic console, controlled by the surgeons, hold a high-magnification camera and surgical instruments to perform minimally invasive, high precision surgery [10]. A systematic review concluded that both ORC and RARC led to similar outcomes in terms of major complications and quality of life [11]. RARC slightly decreased hospital stay and significantly reduced the risk of blood loss [11].

Current literature on the findings of cost-effectiveness of RARC and ORC have been inconsistent [12–14]. Bansal et al. [12] and Smith et al. [14] concluded that the high costs of RARC compared to ORC could be a barrier to cost-effectiveness. Both studies concentrated on cost rather than cost-effectiveness by not including complications or QALYs. A systematic review on comparing RARC to ORC showed that RARC is efficient in yielding fewer complications compared to ORC [15]. Alternatively, Martin et al. [13] and Kukreja et al. [16] found that RARC is cost-effective when accounting for operative time and postoperative care, although the scope of their research is limited due to the small sample size. Martin et al. [13] argued that the postoperative outcomes were more relevant than the cost of the robot and should therefore be taken into account.

Due to high annual costs, NHS England has questioned the routine commission for RARC [18]. Despite this, urologists continue to favour robotic surgery due to its minimally invasive nature [19]. Currently, there are no studies comparing the cost-effectiveness of RARC and ORC in the UK using utilities. Therefore, the aim of this study is to perform a cost-utility analysis (CUA) of RARC versus ORC for bladder cancer treatment, from the perspective of NHS England. This will inform decisions made by NICE regarding the type of surgery to provide in NHS England.

## Methods

A CUA was used to evaluate the two strategies, RARC or ORC for patients with bladder cancer in need of radical cystectomy with an ileal conduit using measures of health-related quality of life (QALY) and costs to calculate the incremental cost-effectiveness ratio (ICER). The data used for modeling was aggregate, anonymised data which was publicly. Thus, no institutional review board approval was required.

### Decision tree and data elements

A decision tree was used to model the outcomes for RARC or ORC (Fig 1). Probabilities, costs and utilities were obtained from a literature review (Table 1) [10, 12, 16, 20]. Probabilities and

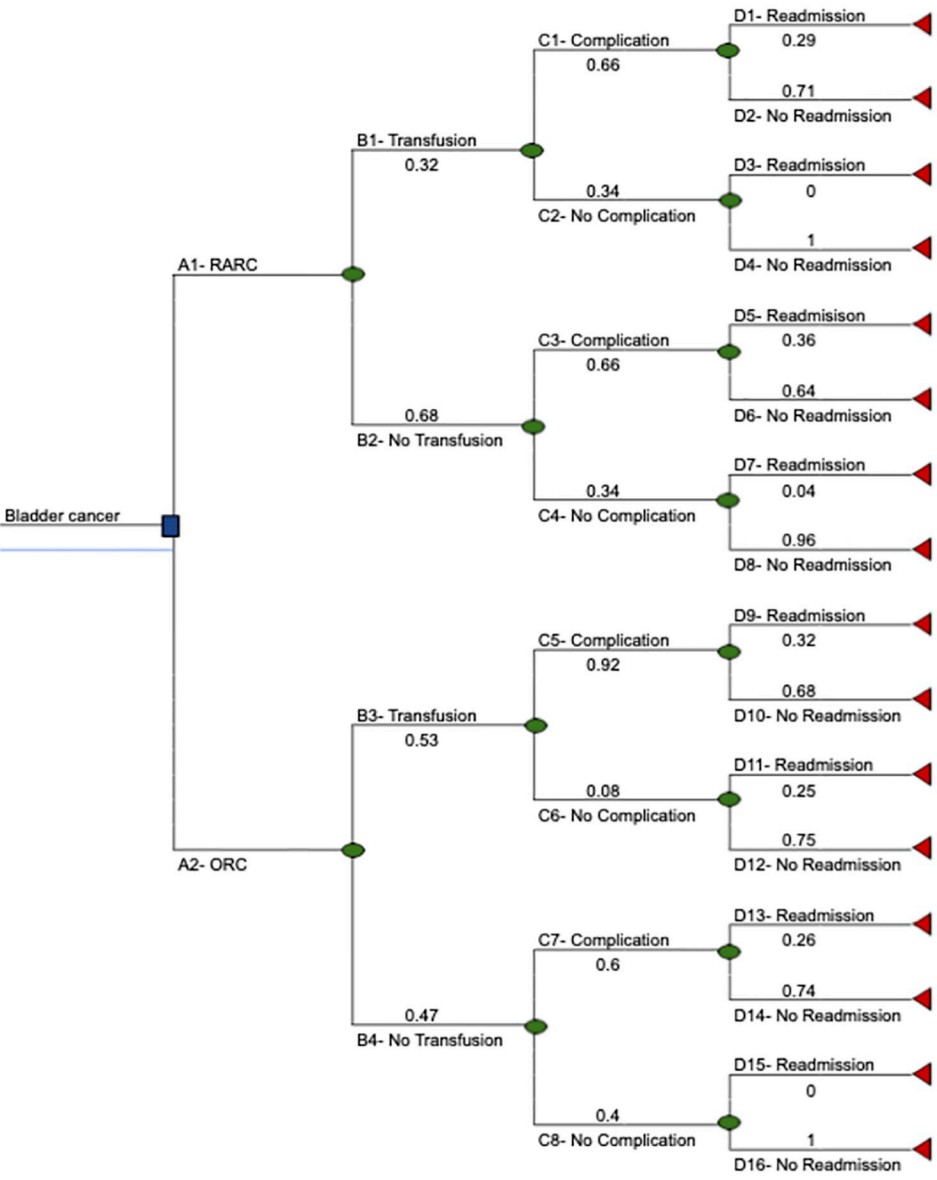

**Fig 1. Three-fold decision tree of RARC and ORC including conditional probabilities of transfusion, complications and readmission.**

**Table 1. Model parameters and range of values for sensitivity analysis: Utilities scores, costs, and probabilities.**

| Probabilities | Base-case value | Univariate sens. analysis | Range | Source |
|---|---|---|---|---|
| B1 | 0.32 | 0.224–0.416 | 0–1 | Kukreja et al., 2020 [16] |
| B2 | 0.68 | | 0–1 | Kukreja et al., 2020 |
| B3 | 0.53 | 0.37–0.689 | 0–1 | Kukreja et al., 2020 |
| B4 | 0.47 | | 0–1 | Kukreja et al., 2020 |
| C1 | 0.66 | 0.462–0.858 | 0–1 | Kukreja et al., 2020 |
| C2 | 0.34 | | 0–1 | Kukreja et al., 2020 |
| C3 | 0.66 | 0.462–0.858 | 0–1 | Kukreja et al., 2020 |
| C4 | 0.34 | | 0–1 | Kukreja et al., 2020 |
| C5 | 0.92 | 0.644–1 | 0–1 | Kukreja et al., 2020 |
| C6 | 0.08 | | 0–1 | Kukreja et al., 2020 |
| C7 | 0.6 | 0.42–0.78 | 0–1 | Kukreja et al., 2020 |
| C8 | 0.4 | | 0–1 | Kukreja et al., 2020 |
| D1 | 0.29 | | 0–1 | Kukreja et al., 2020 |
| D2 | 0.71 | | 0–1 | Kukreja et al., 2020 |
| D3 | 0 | | 0–1 | Kukreja et al., 2020 |
| D4 | 1 | | 0–1 | Kukreja et al., 2020 |
| D5 | 0.36 | | 0–1 | Kukreja et al., 2020 |
| D6 | 0.64 | | 0–1 | Kukreja et al., 2020 |
| D7 | 0.04 | | 0–1 | Kukreja et al., 2020 |
| D8 | 0.96 | | 0–1 | Kukreja et al., 2020 |
| D9 | 0.32 | | 0–1 | Kukreja et al., 2020 |
| D10 | 0.68 | | 0–1 | Kukreja et al., 2020 |
| D11 | 0.25 | | 0–1 | Kukreja et al., 2020 |
| D12 | 0.75 | | 0–1 | Kukreja et al., 2020 |
| D13 | 0.26 | | 0–1 | Kukreja et al., 2020 |
| D14 | 0.74 | | 0–1 | Kukreja et al., 2020 |
| D15 | 0 | | 0–1 | Kukreja et al., 2020 |
| D16 | 1 | | 0–1 | Kukreja et al., 2020 |
| Utilities | Base-case value | Univariate sens. analysis | Range | Source |
| RARC with no complications, readmission or transfusion | 0.8 | 0.6–1 | 0–1 | Kukreja et al., 2020, Sutton et al. 2018 [17] |
| ORC with no transfusions, complications, readmissions | 0.8 | 0.6–1 | 0–1 | Kukreja et al., 2020 |
| Transfusion | -0.1 | -0.05 to -0.3 | 0–1 | Kukreja et al., 2020, Sutton et al. 2018 |
| Short term complication | -0.3 | -0.1 to -0.5 | 0–1 | Kukreja et al., 2020, Sutton et al. 2018 |
| Readmission | -0.1 | -0.005 to -0.3 | 0–1 | Kukreja et al., 2020 |
| Dead | 0 | | 0 | Assumed |
| Costs | Base-case value | Univariate sens. analysis | Range | Source |
| Cost of RARC | 3,794 | 9.656–17,932 | 9.656–17,932 | NHS Tariffs 2018/19 |
| Cost of ORC | 12,004 | 5,805–14,195 | 5,805–14,195 | NHS Tariffs 2018/19 |
| Cost of Follow-up | 227 | | | NHS Tariffs 2018/19 |
| Cost of transfusion | 1,669 | 1,320–2,018 | 1,320–2,018 | NHS Tariffs 2018/19 |
| Cost of complications with readmission | 4,321 | 1,117–6,462 | 1,117–6,462 | NHS Tariffs 2018/19, NICE 2019, Altobelli et al. 2017 |
| Costs of complications without readmission | 216 | 51–280 | 151–280 | NHS Tariffs 2018/19 |
| Costs of readmission without complications | 3,261 | 1,287–5,949 | 1,287–5,949 | NHS Tariffs 2018/19 |

utilities refer to radical cystectomies (RARC and ORC) containing an ileal conduit as a common approach to urinary diversion. Each event after the chance node had a probability conditional to previous events. The sum of the probabilities in branches following one chance node equals 1. Conditional probabilities were calculated using the probabilities in each decision path and were used to calculate the expected cost and QALYs based on input costs and QALYs.

## Costs

Taking an NHS England perspective, the model simulated direct medical expenditures (Table 1). Thus, societal, indirect and individual patient costs such as home medications have not been considered. Resources and costs involved were identified at the aggregate level. Costs for RARC were obtained from Bansal et al. [12]. Most other costs, including preoperative visits, ORC, follow-up of 30/90 days, two blood transfusions, treatment of intervention-related complications, outpatient visits and readmission without complications were obtained from National Cost Collection (NCC) 2018/19 [10]. For healthcare resource groups (HRG) total unit costs were used. Minimal and maximal costs were obtained from selecting the lowest and highest costs of elective, non-elective short and non-elective long stays. Costs that were not found in the NCC were acquired from NICE guidelines and other literature.

The unit costs for ORC, complications requiring no readmission and unit costs for complications requiring admission found in the NCC were averaged. The cost of readmission after none of the above complications included the rate for a regular day or night admission and the rate for readmission for other conditions with intervention. Patients readmitted for adjuvant chemotherapy were not included in the readmission analysis.

Costs were converted to account for inflation. The consumer price index (CPI) was used to calculate inflation and 0.72 was used to convert American dollars to British pounds. Costs were adjusted to January 2021 using the CPI for inflation from the UK Office for National Statistics [21].

## Outcomes

Outcomes were calculated based on different health states including RARC and ORC with no complications, readmission or transfusion, short term complications, transfusion and readmission (Table 1). To be consistent with costs, utilities for patients requiring adjuvant chemotherapy were not included. Utility weights across a 90-day time horizon, ranging from 0–1 were obtained from systematic reports and a meta-analysis [16, 22]. QALYs were calculated using respective conditional probabilities and utility weights. Discounting for QALYs was not assumed to be relevant for this evaluation as the time horizon was less than one year.

## Analysis

QALYs were converted into Net Monetary Benefits (NMB) and Net Health Benefits (NHB). For each intervention, the lower threshold was £20,000/QALY and the upper threshold was £30,000/QALY to show the willingness to pay for an intervention. The ICER was calculated to determine the ratio between the difference in costs and the difference in QALYs of both interventions.

For sensitivity analysis, a threshold analysis including regression models was performed to determine the intervention costs at which the ICER of both interventions met the thresholds of £20,000/QALY, £30,000/QALY and £0/QALY. To account for uncertainty in the parameters and evaluate the impact of various scenarios of the ICER model, a multiple univariate sensitivity analysis was used. Input parameters of the main costs, utilities and probabilities of the main

decision tree nodes (transfusion, complications) were systematically varied by the values found in Table 1. Upper and lower limits for utilities were derived from Kukreja et al. [16]. Ranges for costs were derived from the HRG of the NCC 2018/19 using the lowest and highest costs from elective, non-elective long and short stays [10]. Where variation was not available, 20% variation was applied. Ranges for the probabilities of transfusion and short-term complications were varied by +/- 30%. Based on the varied values, the ICER was recalculated.

## Results

The ICER comparing RARC to ORC yielded £25,526/QALY. This result was plotted on the cost-effectiveness plane as a point estimate on the ICER curve relative to the threshold of £20,000/QALY (Fig 2). The point estimate lay in the top right quadrant above the threshold. The slope of the ICER line was higher than the threshold line.

In comparison with the lower NICE threshold of £20,000/QALY, both types of surgery showed negative NMB, meaning neither surgery is feasible as the costs are higher than the benefits. However, for the higher threshold of £30,000 both values were positive (Fig 3). The value for RARC was higher (£624.61) compared to ORC (£369.09) (Table 2), therefore RARC is worth pursuing here.

Like NMB, for the lower threshold, the NHB for both types of surgery were negative values (Table 2, Fig 4) For the upper threshold, RARC had a value of 0.021, which was higher than the value 0.012 for ORC. This shows RARC is preferred to achieve better health outcomes at £30,000/QALY.

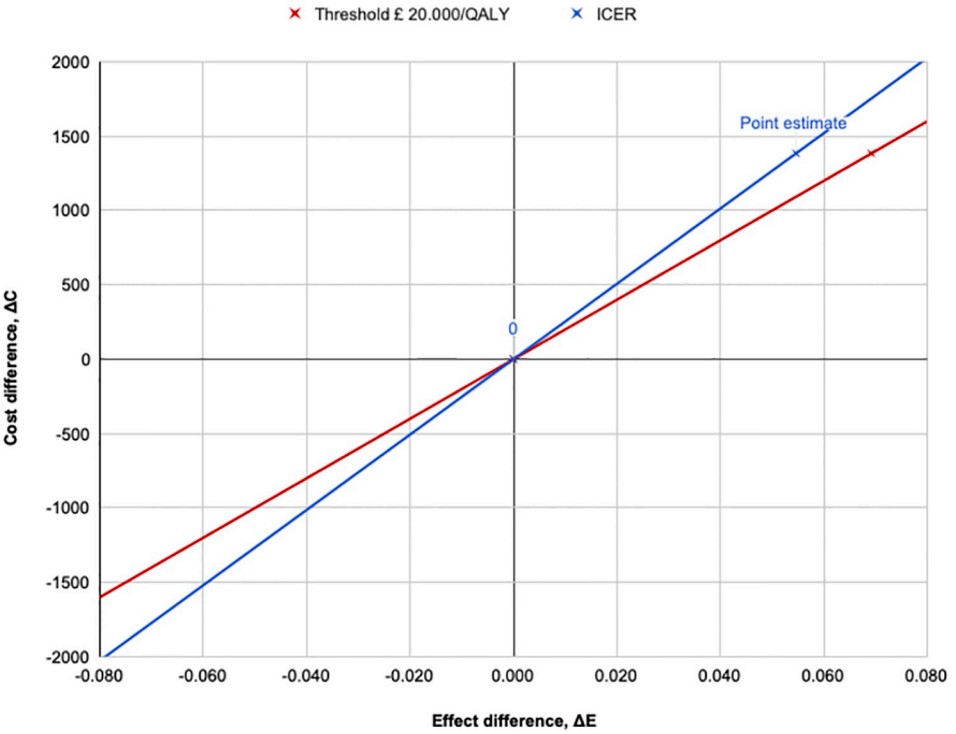

**Fig 2. Cost-effectiveness plane of RARC compared with ORC in relation to the £20,000/QALY threshold.**

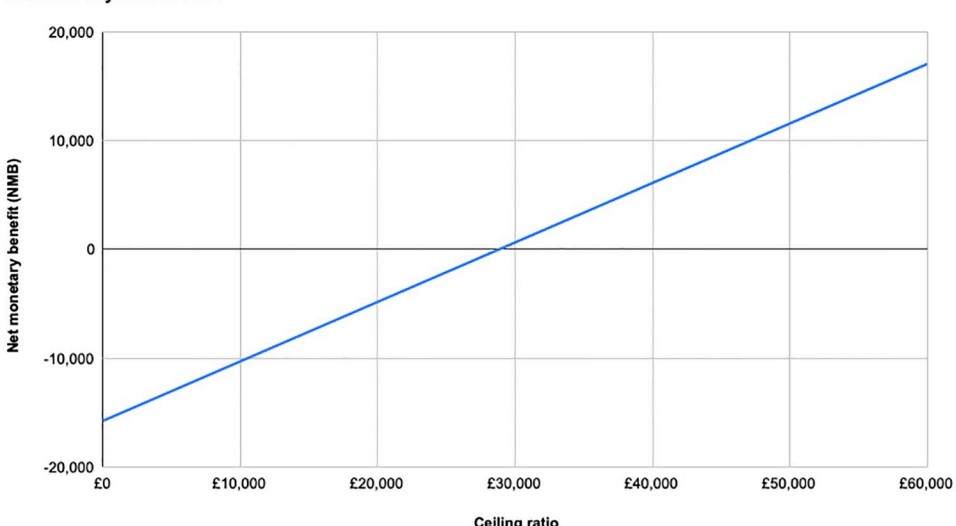

**Fig 3. NMB curve of RARC compared to the ceiling ratio.**

In the threshold analysis, a linear regression model of the ICER for RARC showed that at an intervention cost of £13,497, the £20,000/QALY threshold, and at £14,043, the £30,000/QALY threshold are met (Fig 5). At £12,404 the ICER would be 0. An exponential regression was run for ORC. The ICER meets the £20,000/QALY threshold at £14,788 and the £30,000/QALY threshold at £13,769 in intervention costs.

Multiple univariate sensitivity analysis showed that the ICER was sensitive to variation in the costs of RARC and ORC, the probabilities of the decision tree nodes (B1, B3, C3, C5, C7, B3) and in the utilities of RARC and ORC. The most extreme scenarios yielded dramatically different ICERs, from £101,021 /QALY to -50,363/QALY for variation in RARC costs of +/-30% and from £11,815/QALY to £160,007/QALY for a variation in the probability of complications without prior transfusion (C3) (Table 3). The results were less sensitive to variations in utilities of transfusions, complications, costs of transfusion and all parameters related to readmission. Results were displayed in a tornado diagram (Fig 6).

**Table 2. Base-case results (written to 2 decimal places).**

|  | RARC | ORC | Difference |
|---|---|---|---|
| **Costs (£)** | 15,779.00 | 14,394.66 | 1,384.34 |
| **QALYs** | 0.55 | 0.49 | 0.06 |
| **ICER (£)** |  |  | 25,325.96 |
| **Net Monetary Benefit (£)** |  |  |  |
| Lower | -4,843.32 | -4552.16 |  |
| Upper | 624.61 | 369.09 |  |
| **Net Health Benefit** |  |  |  |
| Lower | -0.24 | -0.23 |  |
| Upper | 0.02 | 0.02 |  |

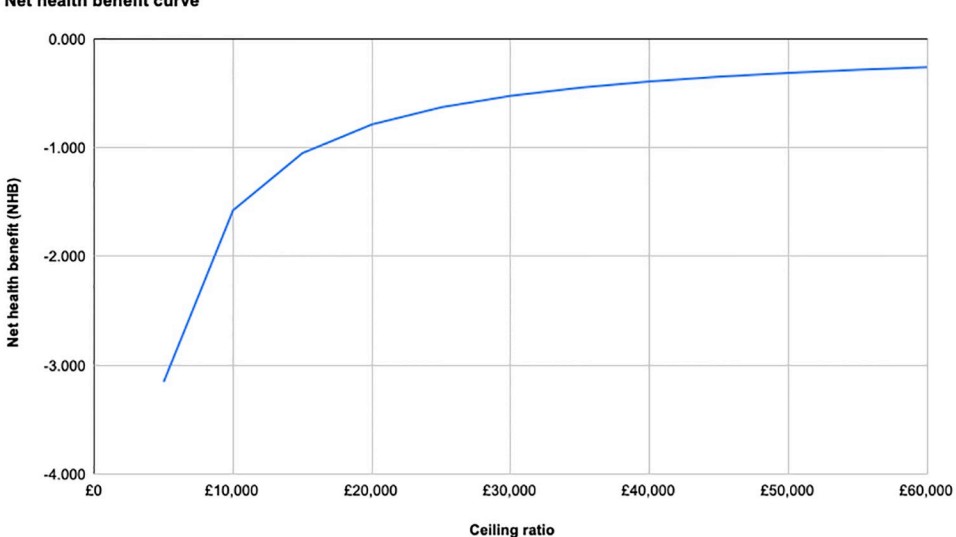

**Fig 4. NHB curve of RARC compared to the ceiling ratio.**

## Discussion

This CUA compared RARC and ORC to determine which is most cost-effective for NHS England. The ICER of £25,526/QALY showed that RARC resulted in an increase of one QALY per £25,526. This ICER was slightly above the NICE threshold of £20,000/QALY making it unlikely to be recommended. Additionally, negative values for the NMB at the lower threshold indicated that the intervention was not feasible from a cost perspective. Regarding the £30,000/QALY threshold, NMB and NHB were positive, indicating that RARC is beneficial. Thus, RARC could be a cost-effective intervention when additional factors were given for justification.

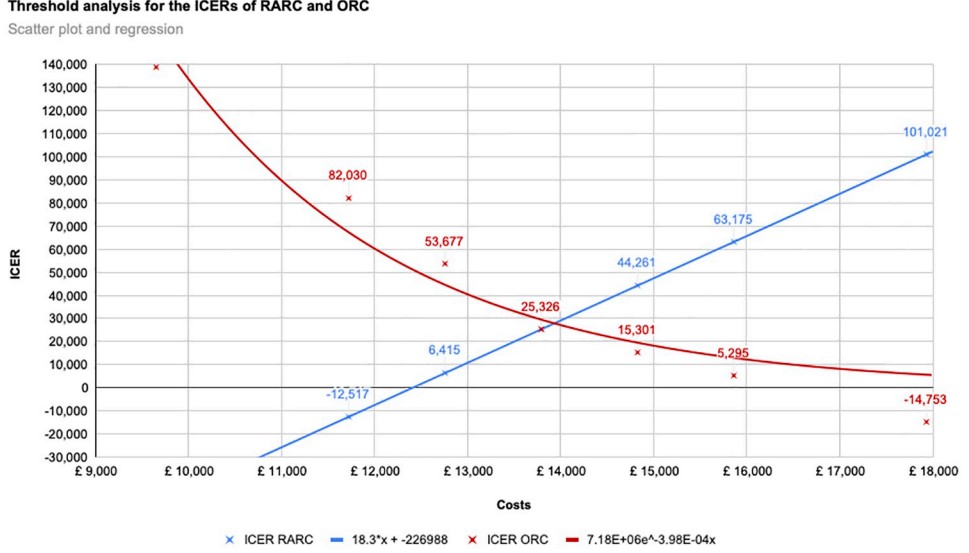

**Fig 5. Threshold analysis for the ICERs of RARC and ORC.**

**Table 3. Multiple univariate sensitivity analysis showing changes of the ICER in relation to variation of the base values of input parameters.**

| Input parameters | Base value | Lower value | Upper value | Lower ICER | Abs Δ lower ICER | Change in % | Upper ICER | Abs Δ upper ICER | Change in % |
|---|---|---|---|---|---|---|---|---|---|
| Cst RARC (9656,17932) | £13,794 | £9,656 | £17,932 | -50,363 | 75,689 | 298.86% | 101,021 | 75,695 | 298.88% |
| Cst ORC (5805,14195) | £12,004 | £5,805 | £14,195 | 138,720 | 113,394 | 447.74% | -14,758 | 40,084 | 158.27% |
| Prob C3 (0.482,0.858) | 0.66 | 2 | 0.858 | 11,815 | 13,511 | 53.35% | 160,007 | 134,681 | 531.79% |
| Prob C5 (0.644,1) | 0.92 | 0.644 | 1 | 152,547 | 127,221 | 502.33% | 20,008 | 5,318 | 21.00% |
| Prob B3 (0.371,0.689) | 0.53 | 0.371 | 0.689 | 83,941 | 58,615 | 231.44% | 11,403 | 13,923 | 54.98% |
| Util RARC (0.6,1) | 0.8 | 0.6 | 1 | -9,527 | 34,853 | 137.62% | 5,437 | 19,889 | 78.53% |
| Util ORC (0.6,1) | 0.8 | 0.6 | 1 | 5,437 | 19,889 | 78.53% | -9,527 | 34,853 | 137.62% |
| Prob C7 (0.42,0.78) | 0.6 | 0.42 | 0.78 | 55,117 | 29,791 | 117.63% | 15,514 | 9,812 | 38.74% |
| Prob C1 (0.462,0.858) | 0.66 | 0.462 | 0.858 | 17,155 | 8,171 | 32.26% | 43,569 | 18,243 | 72.03% |
| Util CX (-0.1,-0.5) | -0.3 | -0.1 | -0.5 | 18,078 | 7,248 | 28.62% | 42,278 | 16,952 | 66.93% |
| Util TF (-0.05,-0.3) | -0.1 | -0.05 | -0.3 | 31,347 | 6,021 | 23.77% | 14,323 | 11,003 | 43.45% |
| Prob B1 (0.224,0.416) | 0.32 | 0.224 | 0.416 | 19,574 | 5,752 | 22.71% | 33,352 | 8,026 | 31.69% |
| Cst TF (1320,2018) | £1,669 | £1,320 | £2,018 | 26,667 | 1,341 | 5.29% | 23,985 | 1,341 | 5.29% |
| Util RA (-0.005,-0.3) | -0.1 | -0.005 | -0.3 | 25,678 | 352 | 1.39% | 24,616 | 710 | 2.80% |
| Cst CX w/ RA (1117,6462) | £4,321 | £1,117 | £6,462 | 25,709 | 383 | 1.51% | 25,070 | 256 | 1.01% |
| Cst CX w/o RA (151,280) | £216 | £151 | £280 | 25,448 | 122 | 0.48% | 25,204 | 122 | 0.48% |
| Cst RA (1287,5949) | £3,261 | £1,287 | £5,949 | 25,253 | 73 | 0.29% | 25,138 | 188 | 0.74% |

NICE has adopted a cost-effectiveness threshold range of £20,000 to £30,000/QALY gained since 1999 [23]. Although ICERs can be used as a decision rule in resource allocation, it could also limit the choices of treatment available to patients. This is due to NICE thresholds not changing for more than 20 years meaning budget and efficiency changes are not accommodated.

The multiple univariate sensitivity analysis showed that variation of one parameter can change the economic conclusion that RARC is not cost-effective by lowering the ICER below the £20,000/QALY threshold. Variables with the greatest impact on the ICER were the costs of RARC and ORC and the changes in the probabilities of transfusion and complications. For instance a small reduction of costs of the RARC from £13,794 to £13,497 would imply an ICER that meets the £20,000/QALY threshold. This highlights the relatively great weight of costs and complications on the ICER compared to health outcomes. Thus, avoiding complications and postoperative transfusions is not only an important quality-of-care outcome for patients but also reduces resources and expenditures.

The explanatory power of univariate sensitivity analyses is limited because, in reality, there are often no isolated changes in parameters, parameters correlate with each other and there is no point of reference for the likelihood of being at different places within the range. More detailed studies are needed to obtain better data on the variation of costs, utilities and probabilities to pin down the ICER and further research to determine the range of parameters required.

Previous studies illustrated that RARC was less cost-effective than ORC, however, these studies did not include health utilities or QALYs [12, 14]. Studies demonstrated that complications were the main source of the cost burden for both RARC and ORC, illustrating the necessity to include these in a cost-effectiveness analysis [13, 24]. Similar to this current study, Kukreja et al. [16] incorporated QALYs and found that without considering QALYs, RARC was notably more expensive than ORC. In addition, they found RARC, compared to ORC, was the more cost-effective option when transfusions and complications were limited [16].

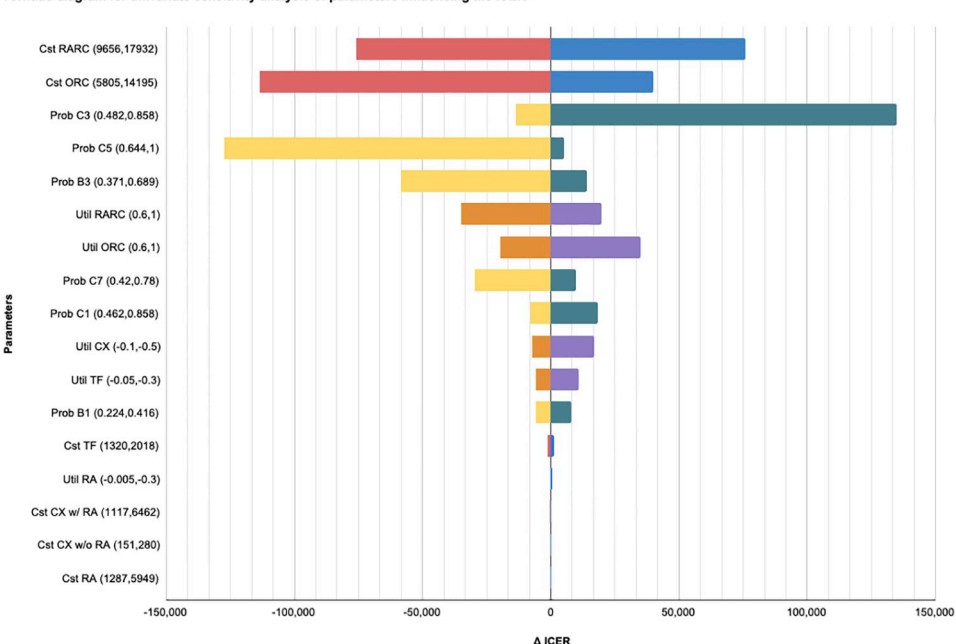

**Fig 6. Tornado diagram representing impact on the ICER when varying one parameter (univariate sensitivity analysis).** Abbreviations: Prob = probabilities; Cst = costs; Util = utilities; TF = transfusion; CX = complications; RA = readmission; w = with; w/o = without.

While previous studies focused on the hospital perspective, this study is conducted through the NHS perspective. Furthermore, this economic evaluation utilised ICER and NMB while previous research used monetary measurements [12, 13].

### Strengths and limitations

To the authors' knowledge, this is currently the most up to date CUA investigating ORC versus RARC in NHS England. The results provide additional evidence to the current debate between the two techniques. Another benefit is the CUA assessed the postoperative complications in detail, which is a novel aspect compared to previous evaluations. This was a vital decision because sensitivity analysis showed that changes in the probabilities of postoperative complications had a great impact on the ICER.

One assumption was that all patients have an equal chance of being offered RARC and ORC. However, some doctors may favour one due to the individual patient's circumstances or lack of available resources. Having these probabilities known with further literature research and addition to the tree would have resulted in more accurate results. The probabilities in the decision tree were based on a similar study completed in the United States [16]. It was assumed that in the UK, patient outcomes will follow the same probabilities. If found, UK-based probabilities would have been used.

The NHS perspective does not account for differences between hospitals related to the quality of care, operative time and patient outcomes such as blood loss, complications and time spent in hospitals [25] affecting overall costs. The quality-of-care patients receive after surgery is unknown. Low quality may lengthen hospital stay, which can increase costs. This variation can occur regardless of the type of surgery the patient has undergone. To estimate these outcomes with increased accuracy, the viewpoint of one hospital could have been taken.

This study did not use operative time as a parameter in the sensitivity analysis as this information is not widely available. If surgery takes longer, the cost for the operating room and surgeons will increase.

Of all patients undergoing RARC, some will be required to stay in the intensive care unit (ICU) after surgery [26]. Complications in this study were broken down into the above-mentioned categories. Costs for ICU treatment were not included as additional probabilities for ICU admission related to those complications could not have been obtained.

As the evidence base in this field increases over time, more information will be available, helping avoid the current limitations. Additionally, having access to a physician working in an NHS England hospital urology department would have made this study more specific and detailed as they would retain knowledge about the pathway of a patient with bladder cancer.

Future research should (1) explore a longer time horizon to establish the cost-effectiveness of RARC and ORC to account for chronic complications, (2) consider specifying the various forms of bladder cancer, such as muscle-invasive bladder cancer, to increase generalisability and (3) consider a micro-costing perspective on an individual hospital basis.

## Conclusion

Based on the results obtained from this CUA, this study cannot recommend RARC over ORC for bladder cancer treatment in England. At the lower threshold, the NMB for both treatments were negative, indicating that the intervention is not feasible from a cost perspective. At the higher threshold, the NMB was higher for RARC compared to ORC meaning the value gained is higher than the net cost. RARC was also associated with a higher NHB hence leads to better health outcomes. This means that at the £30,000/QALY threshold, RARC is more cost-effective for NHS England and could result in an improved utility for patients with bladder cancer. However, the limitations identified need to be overcome with further research to provide further justification for the use of RARC in routine practice to treat bladder cancer patients within NHS England.

## Supporting information

**S1 Dataset. Raw data.**
(PDF)

## Author Contributions

**Conceptualization:** Elena Pizzo.

**Data curation:** Felix Machleid, Jenessa Ho-Wrigley, Ameera Chowdhury.

**Formal analysis:** Felix Machleid, Jenessa Ho-Wrigley, Ameera Chowdhury, Anita Paliah, Ho Lam Poon.

**Methodology:** Felix Machleid, Jenessa Ho-Wrigley.

**Supervision:** Elena Pizzo.

**Writing – original draft:** Felix Machleid, Jenessa Ho-Wrigley, Ameera Chowdhury, Anita Paliah, Ho Lam Poon, Elena Pizzo.

**Writing – review & editing:** Felix Machleid, Jenessa Ho-Wrigley, Ameera Chowdhury, Anita Paliah, Ho Lam Poon, Elena Pizzo.

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
