## [Decision Letter · Decision Letter 0]

15 Mar 2022

PONE-D-21-35203Cost-utility analysis of robotic-assisted radical cystectomy for bladder cancer compared to open radical cystectomy in the United KingdomPLOS ONE

Dear Dr. Wrigley,

Thank you for submitting your manuscript to PLOS ONE. After careful consideration, we feel that it has merit but does not fully meet PLOS ONE’s publication criteria as it currently stands. Therefore, we invite you to submit a revised version of the manuscript that addresses the points raised during the review process.

We look forward to receiving your revised manuscript.

Kind regards,

Isaac Yi Kim, MD, PhD, MBA

Academic Editor

PLOS ONE

Journal Requirements:

2. You indicated that ethical approval was not necessary for your study. We understand that the framework for ethical oversight requirements for studies of this type may differ depending on the setting and we would appreciate some further clarification regarding your research. Could you please provide further details on why your study is exempt from the need for approval and/or confirmation from your institutional review board or research ethics committee (e.g., in the form of a letter or email correspondence) that ethics review was not necessary for this study? Please include a copy of the correspondence as an ""Other"" file.

Unfunded studies

No authors have competing interests

6. Please note that in order to use the direct billing option the corresponding author must be affiliated with the chosen institute. Please either amend your manuscript to change the affiliation or corresponding author, or email us at plosone@plos.org with a request to remove this option.

Reviewers' comments:

Reviewer's Responses to Questions

**Comments to the Author**

1. Is the manuscript technically sound, and do the data support the conclusions?

Reviewer #1: Yes

2. Has the statistical analysis been performed appropriately and rigorously? 

Reviewer #1: Yes

3. Have the authors made all data underlying the findings in their manuscript fully available?

Reviewer #1: Yes

4. Is the manuscript presented in an intelligible fashion and written in standard English?

Reviewer #1: Yes

5. Review Comments to the Author

Reviewer #1: The ICER for robotic versus open cystectomy was close, although did not meet the threshold of 20,000 pound/QALY. Given how close it is and the finding that RARC did improve QALY, I would suggest also exploring what modifications would have to exist to reach this level.

Were choice and approach of urinary diversion accounted for? Many of the RARC studies included an open incision for the urinary diversion. What are the implications for an intracorporeal neobladder or ileal conduit?

6. PLOS authors have the option to publish the peer review history of their article (what does this mean?). If published, this will include your full peer review and any attached files.

Reviewer #1: No

---

## [Author Response · Author response to Decision Letter 0]

1 Jun 2022

Reviewer Comment 1: The ICER for robotic versus open cystectomy was close, although did not meet the threshold of 20,000 pound/QALY. Given how close it is and the finding that RARC did improve QALY, I would suggest also exploring what modifications would have to exist to reach this level.

Response: We thank you for this suggestion. Based on the input parameters (probabilities, benefits and costs), the sensitivity analysis and tornado diagram showed that the ICER was mainly influenced by the cost of RARC (see figure 6). On this basis, the linear regression model of the ICER for RARC showed that if the cost of the intervention was reduced to £13.497, the threshold of £20,000/QALY would be reached (see figure 5). We added one sentence in the discussion of the results to highlight the modification of costs to make the ICER reach this level. 

Reviewer Comment 2: Were choice and approach of urinary diversion accounted for? Many of the RARC studies included an open incision for the urinary diversion. What are the implications for an intracorporeal neobladder or ileal conduit?

Response: Thank you for raising this additional point. As for the parameters used to model the ICER, the probabilities and utilities (based mainly on Kukreja et al. 2018) refer to radical cystectomies (RARC and ORC) containing an ileal conduit for urinary diversion (see Kukreja et al. 2018). We included an explanatory sentence in the methods section to reflect on this. We could not find data on probabilities and utilities that refer to the implications of an intracorporeal neobladder.

Reviewer:1. We would like to apologies for the delays on your mansucript. Thank you for providing the email thread to confirm that your study does not require IRB approval as pubicility available de-identified data was used. To ensure transparency in reporting, we would be grateful if you could also provide a statement in the methods section of the manuscript text indicating the above.

Response: To ensure transparency in reporting, we would be grateful if you could also provide a statement in the methods section of the manuscript text indicating the above.

The data used for modeling was aggregate, anonymised data which was publicly. Thus, no institutional review board approval was required.

---

## [Editor Report · Decision Letter 1]

9 Jun 2022

Cost-utility analysis of robotic-assisted radical cystectomy for bladder cancer compared to open radical cystectomy in the United Kingdom

PONE-D-21-35203R1

Dear Dr. Wrigley,

We’re pleased to inform you that your manuscript has been judged scientifically suitable for publication and will be formally accepted for publication once it meets all outstanding technical requirements.

Kind regards,

Isaac Yi Kim, MD, PhD, MBA

Academic Editor

PLOS ONE
---

## [Editor Report · Acceptance letter]

2 Sep 2022

PONE-D-21-35203R1 

Cost-utility analysis of robotic-assisted radical cystectomy for bladder cancer compared to open radical cystectomy in the United Kingdom 

Dear Dr. Ho-Wrigley:

I'm pleased to inform you that your manuscript has been deemed suitable for publication in PLOS ONE. Congratulations! Your manuscript is now with our production department. 

Kind regards, 

on behalf of

Dr. Isaac Yi Kim 

Academic Editor

PLOS ONE